# Open Plant Phenotype Database of Common Weeds in Denmark

**Simon Leminen Madsen** [1],* , **Solvejg Kopp Mathiassen** [2] , **Mads Dyrmann** [3] ,
**Morten Stigaard Laursen** [1] , **Laura-Carlota Paz** [4] and **Rasmus Nyholm Jørgensen** [1]

1   Department of Engineering, Aarhus University, DK-8200 Aarhus N, Denmark; msl@eng.au.dk (M.S.L.);
    rnj@eng.au.dk (R.N.J.)
2   Department of Agroecology, Aarhus University, DK-4200 Slagelse, Denmark; sma@agro.au.dk
3   School of Engineering, Aarhus University, DK-8200 Aarhus N, Denmark; madsdyrmann@ase.au.dk
4   I·GIS A/S, DK-8240 Risskov, Denmark; lcp@i-gis.dk
*   Correspondence: slm@eng.au.dk

**Abstract:** For decades, significant effort has been put into the development of plant detection and classification algorithms. However, it has been difficult to compare the performance of the different algorithms, due to the lack of a common testbed, such as a public available annotated reference dataset. In this paper, we present the Open Plant Phenotype Database (OPPD), a public dataset for plant detection and plant classification. The dataset contains 7590 RGB images of 47 plant species. Each species is cultivated under three different growth conditions, to provide a high degree of diversity in terms of visual appearance. The images are collected at the semifield area at Aarhus University, Research Centre Flakkebjerg, Denmark, using a customized data acquisition platform that provides well-illuminated images with a ground resolution of ∼6.6 px mm$^{-1}$. All images are annotated with plant species using the EPPO encoding system, bounding box annotations for detection and extraction of individual plants, applied growth conditions and time passed since seeding. Additionally, the individual plants have been tracked temporally and given unique IDs. The dataset is accompanied by two experiments for: (1) plant instance detection and (2) plant species classification. The experiments introduce evaluation metrics and methods for the two tasks and provide baselines for future work on the data.

**Keywords:** dataset; plant phenotyping; plant seedlings; weed control

## 1. Introduction

Visual recognition systems are becoming increasingly widespread for farm management in modern agriculture [1–4]. The systems are typically used in conjunction with remote sensing technologies to extract knowledge about various field conditions. Airborne sensing platforms, such as satellites and Unmanned Aerial Vehicles (UAVs), have in recent years been demonstrated to be efficient for mapping vegetation coverage [5] and weed infestations (binary distinction between weed or crop) [6–8] in fields, due to the relatively high capacity of such systems. However, the high capacity of airborne platforms usually comes at the cost of spatial resolution, which together variable scene illuminations causes a high risk of erroneous plant detections and identifications, which might lead to faulty syllogism/interpretations [9,10]. Low spatial resolutions also impair the capability of aerial sensing systems, to distinguish between individual plant objects and to perform plant species classification, especially at the early growth stages of the plants [6,10–12]. Alternatively, ground-based sensing platforms (proximal sensing) generally provide higher spatial resolution compared to airborne platforms, which can be used to efficiently map the population and composition of plants in fields at



species level [10,12–14]. Proximal sensing systems can also be mounted directly on the farming equipment, which is a highly desired property, due to an increasing focus on automation (e.g., agricultural robots) of the agricultural practices [3,13,15]. Detailed plant population maps can, for instance, be used in decision support systems for weed management, to optimize control strategies according to the observed weed species and composition, which can provide significant reductions in herbicide usages [3,12,16,17]. Proximal sensing and visual recognition can also be applied for other applications, such as phenotyping, diseases and pests detection, grain quality and plant fitness estimation, etc. [2,18].

Visual recognition of individual plants is generally a challenging task, as plant appearances depend heavily on several factors such as: species, temporal changes and environmental conditions (soil type, nutrients, climate, etc.) [15,19,20]. Historically, algorithms for visual recognition of plants have been based on the empirical measurement and handcrafted features (e.g., shape and texture descriptors, Fourier descriptors, or active shape modeling), which were processed using relatively simple discriminative models such as linear classifiers or support vector machines [18,21]. Newer methods for visual recognition of plants are primarily inspired by a more data-driven approach such as deep learning (DL) [1,3], which have shown impressive results in various other domains. E.g., Dyrmann et al. [22,23] has demonstrated effective detection and classification of weeds in highly occluded cereal fields using convolutional neural networks, which was used to map weed populations. Additionally, the systems demonstrated by Lottes et al. [24] can pixel-wise segment weed and crop in images based on a time series of images, which enables them to treat plants individually. Zhang et al. [4] provide a comprehensive review of visual recognition systems for dense scenes analysis in agriculture, which concludes "*By comparing the DL methods with other methods in the survey paper, it shows that DL provides better performance in dense scenes and is superior to other popular image processing techniques.*"

DL algorithms are primarily trained by applying supervised learning, which requires large quantities of annotated training data [25,26]. However, for visual recognition of plants, the availability of such training data is somewhat limited, as there only exist few public datasets. Furthermore, these datasets only represent relatively few plant species, growth condition or growth seasons, which reduces the inter-species and intra-species variability with respect to the plant appearances, thus making them imperfect for building robust DL systems. Due to these limitations, research in this domain is often performed on private datasets, which makes it difficult to validate and compare results from different studies/projects. To overcome these issues and boost research in visual recognition of plants more public available datasets are needed [1,27,28].

The aim of this study is to develop a new public dataset for visual recognition of plants. This dataset should include several different plant species and a high degree of intra-species variability with respect to plant appearances, as these properties are currently lacking/limited in existing public datasets.

This work presents the Open Plant Phenotype Database (OPPD), a large-scale dataset for visual recognition of plant seedlings. The dataset provides high resolution top-down RGB images of plants, cultivated at the semifield area at Aarhus University, Research Centre Flakkebjerg, Denmark. The dataset consists of 7590 images with 315,038 plant objects, representing 64,292 individual plants from 47 different species. Each plant species has been cultivated using three growth conditions (ideal, drought and natural) and tracked temporally to achieve high intra-species variability. This work also introduces two baseline experiments, which introduces two use-cases of OPPD: (1) plant instance detection and (2) plant species classification. The baseline experiments are primarily used to introduce the evaluation method and metrics.

The remainder of the manuscript is organized as follows: Section 2 provides a description of the data collection and annotation process. Section 3 provides an overview of the dataset content. Section 4 provides a description and results from the baseline experiments. Section 5 and 6 provides discussion and conclusion.

The full dataset are available at: https://vision.eng.au.dk/open-plant-phenotyping-database/.

*Related Work*

Most closely related to this work is the Plant Phenotyping Datasets (PPD) by Minervini et al. [29,30] and the Plant Seedlings Dataset (PSD) by Giselsson et al. [31]. Both datasets consist of RGB images representing plant seedlings cultivated in an indoor environment. PPD provides samples from 123 individual plants representing two rosette species, which are annotated with segmentation masks and bounding boxes for full plants and individual leaves. PSD provides samples from ∼960 individual plants representing twelve different species, which are annotated with bounding boxes and segmentation mask. In both datasets, the plants have been tracked temporally over the course of several weeks to monitor the temporal changes in appearance. For PPD, different growth conditions were applied to a limited degree, as a subset of the plants also were part of a watering experiment.

Other datasets are collected in a real field environment, where the data represents natural growing plants. Examples of this are the Sugar Beets Dataset (SBD) by Chebrolu et al. [32], the DeepWeed dataset by Olses et al. [33], and the Leaf Counting Dataset (LCD) by Teimouri et al. [34]. SBD is a multi-modal dataset consisting of multi-spectral and RBG-D images and laser scans, collected 2–3 times a week for three months. The SBD provides ∼300 images, that have been pixel-wise annotated as vegetation, belonging to 10 different plant species (Sugar beets and nine weeds) or as non-vegetation. The DeepWeeds dataset consists of ∼17,500 RGB images of eight significant weed species in the north–east Australia. The images are collected in-situ under natural light including several factors of variation, namely: illumination, rotation, scale, focus, occlusion, dynamic backgrounds; as well as geographical (Australia) and seasonal variation. LCD consists of ∼12,000 individual plant cut-outs representing 18 different weed species. The images of LCD are collected in various fields around Denmark, thus representing several different growth conditions and growth stages.

## 2. Methodology

### 2.1. Data Collection

The data collection was performed at Research Centre Flakkebjerg, Aarhus University (N55°19′28.4736″, E11°23′24.0144″), where several plant species were cultivated in a semi-field setting to mimic natural growth conditions. The plant species were selected to represent the 46 most common monocotyledon (grass) and dicotyledon (broadleaved) weeds species in arable crops in Denmark.

At the vegetative stages, the main characteristics used for identification of the plant species are the size, leaf count and shape of the cotyledons and true leaves. Other characteristics are the leaf color, leaf surface (waxy, hairy) and leaf position (angle-to-stem). However, it is well known that several of these traits are affected by environmental factors. For example at growth conditions with low soil moisture, the leaves are smaller and more dark green with a more developed wax layer on the surface, while low nutrient availability often pose leaves and stems to turn yellow or red and cause necrotic spots on the leaves [15,19,20]. In order to provide a broad spectrum of the plant appearances, our plant species were cultivated in polystyrene boxes (0.40 × 0.40 × 0.15 m) under three different controlled growth conditions (partly inducing stress). The different growth conditions were, G1: a potting mixture consisting of a sandy loam soil, sand and peat (2 : 1 : 1 $w/w$) including all necessary micro- and macro-nutrients, optimum soil moisture, G2: a sandy loam soil with optimum fertilizer supply and sub-optimum watering and G3: a sandy soil with low nutrient content and optimum soil moisture. The growth boxes were initially watered to field capacity and seeds were placed at the soil surface and covered with ∼1 cm soil. Subsequently, the boxes were placed on outdoor tables (see Figure 1a) and watered from below three times a day; occasionally a light shower was applied on top to keep the surface moist. After seedling emergence, the plants were thinned to minimize overlapping plant leaves. For the sandy loam soil (condition G2) drought stress was induced in the plants by stopping the watering when the seedling had 2 true leaves. Natural rain was avoided using an automated rain

cover that only was activated when rain was detected by nearby rain sensors. All seeds applied in the experiments were provided by the seed bank at Research Centre Flakkebjerg, Aarhus University.

### 2.1.1. Imaging Acquisition Setup

Image acquisition was performed by the setup described in Madsen et al. [35]. The setup comprised of an imaging system, which was mounted on a railing system above the seedbeds to provide a top-down image of the plants. Figure 1 shows the full data acquisition setup.

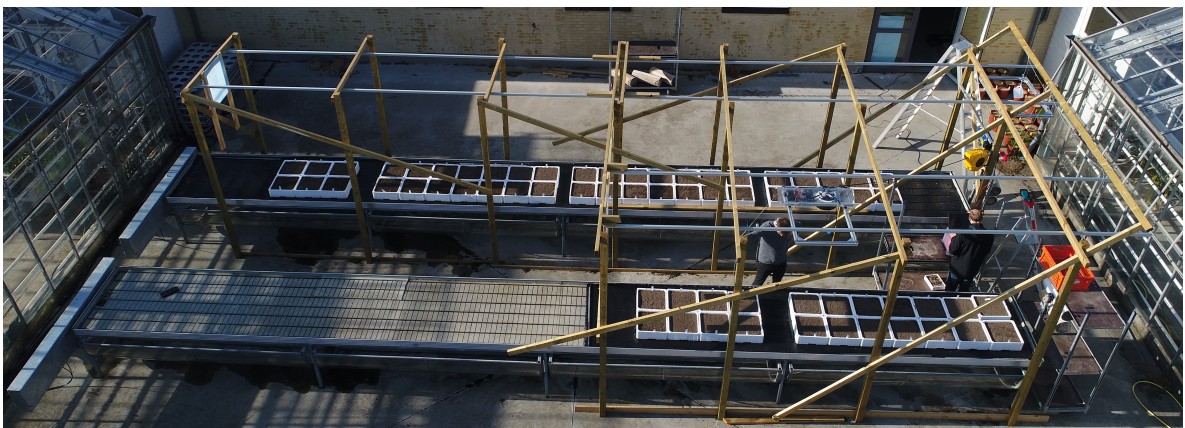

**(a)**

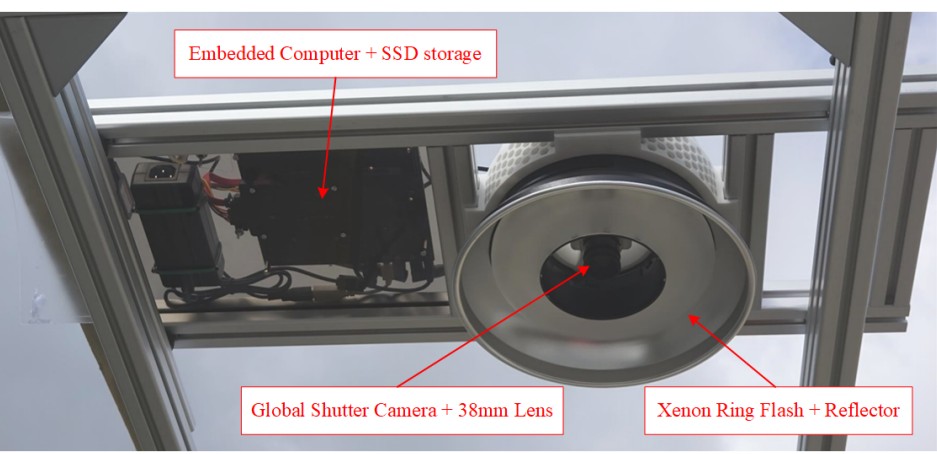

**(b)**

**Figure 1.** Data acquisition setup. (**a**) Bird's-eye view of the data acquisition setup. The image shows the polystyrene boxes placed on the outdoor test tables, while the imaging system mounted on the railing system during a test of the system. The test tables rain covers are stored away under the tables when rain is absent. (**b**) Frog's-eye view of the imaging system.

The imaging system itself was an updated version of the one applied for in-situ data acquisition by Laursen et al. [14]. The imaging system is shown in Figure 1b The system consisted of a 12.3 Mpix global shutter camera (Flir GS3-U3-123S6C) and a 38mm lens at F/4 (Schneider Kreuznach Xenon-Topaz XN 2,0/38-0901). The images were illuminated by a Xenon ring flash (Paul C. Buff, AlienBee ABR800) with the lens mounted in the center of the ring. This setup was used to provide well illuminated, high-resolution images. The imaging system was mounted 1.7 m above the seedbeds to provide a ground field of view of $0.62 \times 0.45$ m, which resulted in a ground resolution of $\sim 6.6$ px mm$^{-1}$. The imaging acquisition system was controlled by an Nvidia TX2 based computer with a 500 GB SSD for data storage. The image acquisition was triggered for every 0.15 m displacement along the railing system, which was measured by a laser distance sensor (Lightware SF11/C).

2.1.2. Recording Procedure and Image Development

Recording was performed by traversing the imaging system above each of the polystyrene boxes 1–3 times a day, dependent on the season of the trial. The plants from each trial were tracked over a development period from seedling emergence to the 6 to 8 leaf stage (from 36 to 60 days) representing the relevant stages for weed control in the field. Data collection was performed over four trial seasons: 2017spring, 2017autumn, 2018summer and 2019summer. The collected images were stored in a 16 bits/px Bayer format, to save space on the acquisition computer and avoid compression/interpolation caused data loss. The raw images were converted to RGB by applying the Malvar He Cutler debayer algorithm [36] post data collection. Additionally, the RGB images were preprocessed by applying corrections to white-balance and gamma, to achieve natural appearing colours in the images.

*2.2. Data Annotation*

Each image showed ∼1.5 full polystyrene boxes as the camera field of view was slightly larger than the area of a single polystyrene box (0.40 × 0.40 m). Thus, each image was annotated with plant species and growth conditions according to the polystyrene box that was most visible within the field of view of the corresponding image. Additionally, the corners of this primary polystyrene box are annotated, by applying a square detection algorithm that found for the best fitting square in the image with an area similar to the inside of the box, see Figure 2a. Due to the camera's trigger rate several images from the same data collection run, could represent the same primary polystyrene box. To remove these "duplicates" in the dataset, only the image where most of the polystyrene box is visible, was selected for further processing. Only the visual content within each primary polystyrene box was of interest for each image, as potential neighboring boxes will be the primary box in another image. Thus, all image content outside the primary box was considered noise and removed from the images, see Figure 2b.

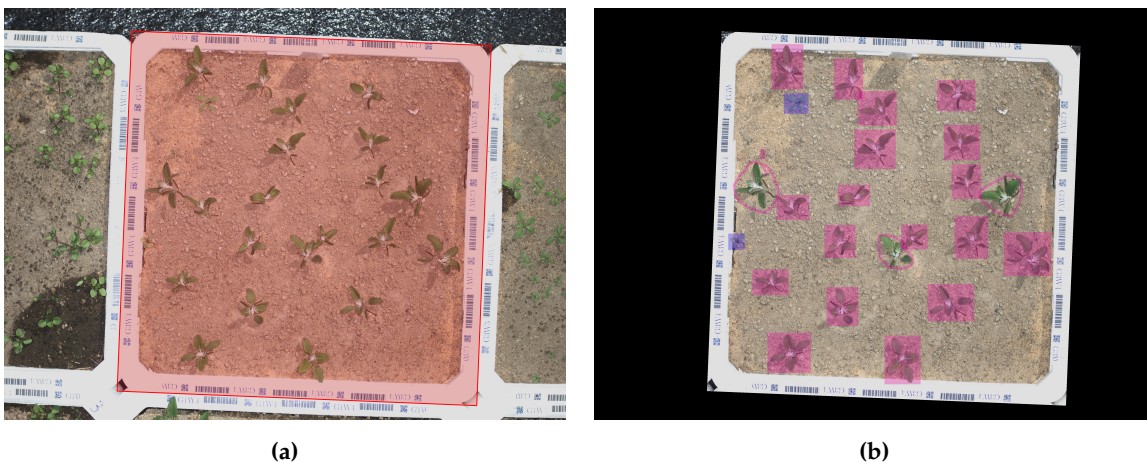

(**a**)　　　　　　　　　　　　　　　　　　　　　　(**b**)

**Figure 2.** Annotation of images. (**a**) Annotation of primary polystyrene box in full RGB image. (**b**) Annotation of individual plants in the RGB image after removing all content outside the primary polystyrene box. The bounding boxes are annotated in RoboWeedMaPS's online annotation tool. The rectangular annotation corresponds to the assisting annotation algorithm's region proposals, while the three hand drawn annotations corresponds to manual corrections performed by a human annotator. The colours of the bounding boxes corresponds to different plant species.

The individual plants in each image were annotated using the RoboWeedMaPS online tool (I·GIS A/S, Risskov, Denmark), which provided a machine learning assisted annotation framework for bounding box and species annotation of plants based on [22]. The RoboWeedMaPS annotation tool provided region proposals and preliminary classification into either monocotyledon or dicotyledon for

all plants in a given image. Thus, the manual annotator only needed to validate the proposals and correct potential errors. An example of the annotated bounding boxes can be seen in Figure 2b.

Additionally, the bounding boxes were labeled with plant species by using the European and Mediterranean Plant Protection Organization (EPPO) encoding system, which provides support for labels at multiple taxonomic levels (https://gd.eppo.int/). The individual bounding boxes were labeled at the highest recognizable taxonomic level by using the RoboWeedMaPS's correction tool feature, which allowed the annotator to inspect all annotated plants across multiple images and group these together by species. This feature also enabled the annotator to easily recognize outliers from other species. In several instances, the plants grew beyond the box and became visible in the neighboring box. In these cases, the objects were also bounding box annotated, but only labeled as either monocotyledon or dicotyledon. The same applied to all plant objects which clearly were not the sown species and are hence a weed within the sown species as illustrated in Figure 3.

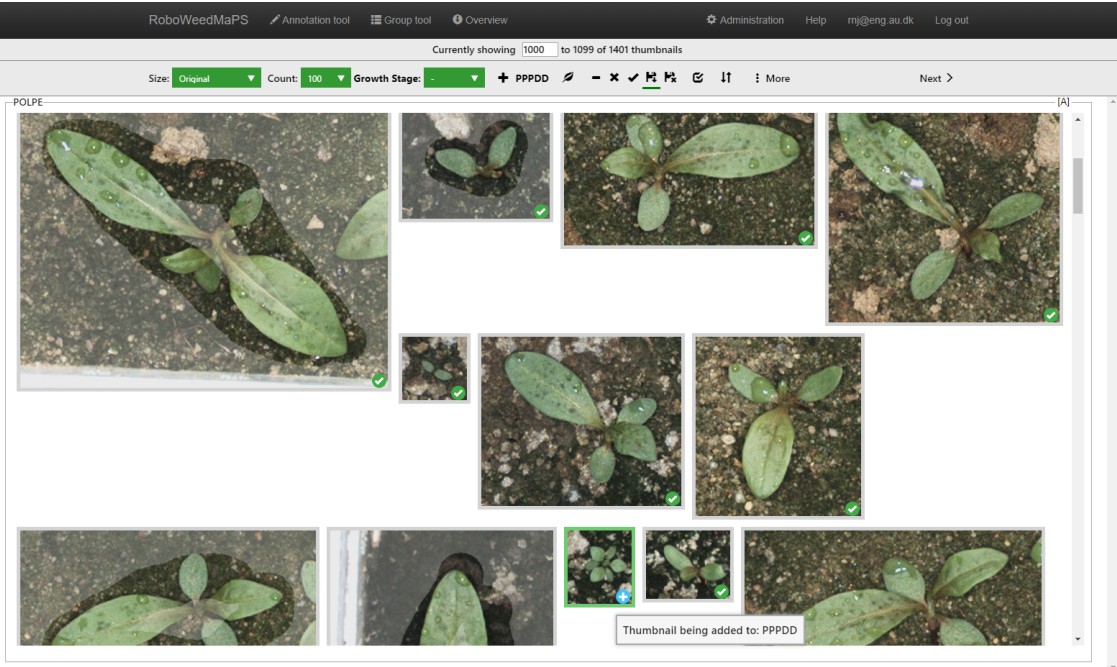

**Figure 3.** RoboWeedMaps's correction tool feature used to manually verify the classification of POLPE (Persicaria maculosa, Redshank). The object with a white cross in a blue circle icon is being relabeled as PPPDD (dicotyledonous plants) class. Notice some objects are shaded as these are manually annotated, to correct for lacking or incorrect automatic bounding box annotation.

An automated algorithm was used to track the individual plant/bounding box annotations temporarily. For each image, the polystyrene box corners was used to calculate an affine transformation into a common coordinate system (axis-aligned with a fixed size of $1000 \times 1000$ pixels). Bounding boxes in two temporally succeeding images was treated as a match if the Intersection over Union (IoU) between them in the common coordinate system is >0.3, which corresponds to 50% overlap (this threshold value was determined empirically). The steps of the algorithm are visualized in Figure 4. In the case of multiple potential matches, the best pairwise matching was computed using the Hungarian algorithm Kuhn [37] with $1 - IoU$ as the cost matrix.

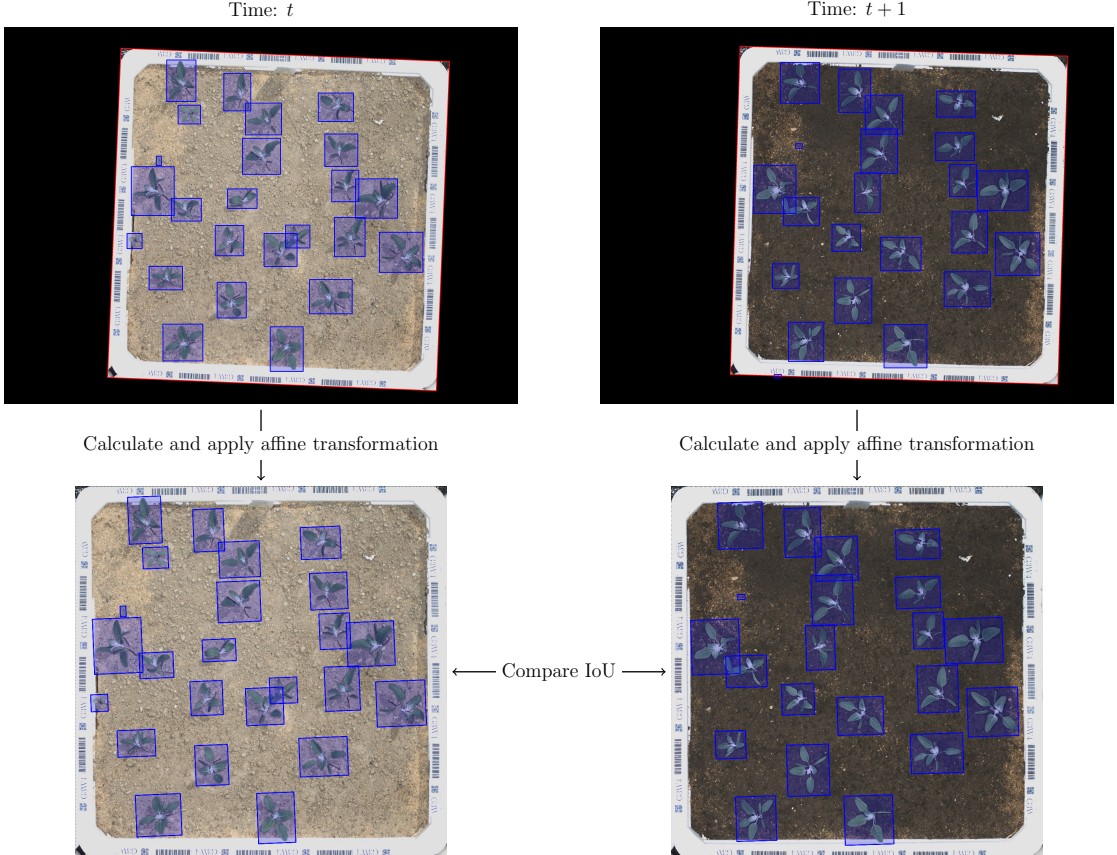

**Figure 4.** Matching bounding boxes example. For each image, an affine transformation was calculated to map the boxes into a common coordinate system. The bounding boxes in two temporal succeeding frames was matched if the Intersection over Union (IoU) between them were >0.3.

## 3. Dataset Content

The OPPD consists of 7590 images with $315,041$ plant objects, representing $64,292$ individual plants from 47 different species. (Note: only 46, species were sown, but the seeds from SONOL had been mixed with SONAS seeds, resulting in a total of 47 species.)

The content of the dataset is divided into two parts: Full box images and individual plant cut-outs (plant objects). Both parts are described in the following subsections.

### 3.1. Full Box Images

The full box images represent individual polystyrene boxes. The distribution of images and bounding boxes for each sown species are summarised in Table A1. Each full box image is annotated with the following general information:

- Date: When the image was recorded
- Trial id: Trial the image belongs to
- Box id: Identifier for the polystyrene box in the specific trial
- Growth condition: Applied growth condition
- Image id: Unique identifier for the image

Additionally, any plants in the full box image are annotated with:

- EPPO code: Class of the plant, annotated at the highest recognizable taxonomic level.
- Bounding box: Pixel coordinates for location (xmin, ymin, xmax, ymax)
- Bounding box id: Unique identifier for the bounding box

- Plant id: Unique identifier for the plant, assigned using the automated plant tracking

## 3.2. Individual Plant Cut-Outs

The individual plant cut-outs are square image cut-outs corresponding to all annotated bounding boxes in the full images. The square crop is determined by the largest dimension of each bounding box. The distribution of plant cut-outs and individual plants for each species are summarised in Table A2. Each plant cut-out is annotated with the following information:

- Date: When the image was recorded
- EPPO code: Class of the plant, annotated at the highest recognizable taxonomic level.
- Trial id: Trial the image belongs to
- Box id: Identifier for the polystyrene box in the specific trial
- Growth condition: Applied growth condition
- Bounding box id: Unique identifier for the bounding box
- Plant id: Unique identifier for the plant, assigned using the automated plant tracking
- Source image id: Unique identifier for the image

## 4. Baseline Experiments

To introduce use cases for the dataset this section presents two experiments to form a baseline for future comparisons. The experiments cover two tasks: a plant instance detection task and a plant classification task. The main scope of these experiments is to introduce the expected evaluation methods and metrics, thus the applied models are simply "out-of-the-box" implementations, so the results can easily be replicated.

### 4.1. Plant Instance Detection

The goal of the plant instance detection task is to train and test an algorithm to determine the locations of all plants in the full box images. Each plant location is to be detected with the minimum axis-aligned bounding box, as annotated in the dataset.

As only the plant locations, and not the plant species are of interest for this task, all bounding boxes in the full box images are relabeled to: 1PLAK (EPPO code for the kingdom of plants).

#### 4.1.1. Evaluation Method

To validate how well the plant instance detection algorithm generalizes, a 10-fold cross-validation scheme is applied. Thereby, the results are not biased due to a specific split of training and validation data. The full box images are divided into ten splits, based on trail id and box id, so each polystyrene box is only present in a single data split. Additionally, boxes from the same species are put in separate splits. This ensures a clear separation in training and validation data for each fold.

As in the COCO object detection task [38], the results are reported using the average precision (AP) and the average recall (AR) metrics.

AP is calculated as the interpolation of the precision-recall curve [39]. Similar to the COCO object detection task, AP is in this instance calculated using a 101-point interpolation [38]:

$$AP = \frac{1}{101} \sum_{r \in \{0, 0.01, \dots, 1\}} p_{interp}(r) \qquad with \quad p_{interp}(r) = \max_{\tilde{r}: \tilde{r} \geq r} p(\tilde{r}), \tag{1}$$

where $p(\tilde{r})$ is the measured precision at recall $\tilde{r}$ [39]. The precision-recall curve depends on the IoU between the ground truth bounding boxes and the predicted bounding boxes, which defines whether a plant is detected. In this work, AP is reported as the average over multiple IoU detection thresholds ($AP^{IoU=.50:.05:.95}$), with a detection threshold of $IoU > 0.5$ ($AP^{IoU=.50}$), and with a detection threshold of $IoU > 0.75$ ($AP^{IoU=.75}$).

AR is the maximum recall given a fixed number of detections per image as defined by Hosang et al. [40]:

$$AR = \frac{2}{n} \sum_{i}^{n} \max(IoU(gt_i) - 0.5, 0), \tag{2}$$

where $IoU(gt_i)$ is the IoU between the annotation $gt_i$ and the closest detection proposal, and $n$ is the number of detections per image [40]. As in the COCO detection task, AR is reported for max 1, 10 and 100 detections per image [38].

### 4.1.2. Baseline

The baseline detection model is based on the TensorFlow implementation [41] of faster RCNN [42] with ResNet50-v1 [43] as feature extractor. The model parameters are initialized with weights pre-trained on the COCO dataset [38]. Additionally for training the model on OPPD, the detection model is configured with the default training parameters as used in TensorFlow's example implementation for the COCO object detection task. The baseline model is evaluated using the above-mentioned evaluation approach. Each model is trained for 400,000 steps on the training data and then evaluated on the evaluation data for each split. The number of training iterations are determined empirically. The results for the baseline models are summarized in Table 1.

**Table 1.** Results from the plant instance detection baseline. The reported values are the mean and standard deviation over all ten splits in the 10-fold cross validation scheme.

| Metric | Score |
|---|---|
| $AP^{IoU=.50:.05:.95}$ | $37.01 \pm 2.43$ |
| $AP^{IoU=.50}$ | $65.53 \pm 4.54$ |
| $AP^{IoU=.75}$ | $37.30 \pm 2.64$ |
| $AR^{n=1}$ | $1.97 \pm 0.23$ |
| $AR^{n=10}$ | $15.34 \pm 2.15$ |
| $AR^{n=100}$ | $44.09 \pm 2.18$ |

### *4.2. Plant Species Classification*

The goal of the plant species classification task is to train and test an algorithm to classify the individual plant cut-outs, i.e., the content of the bounding boxes. In this task, only samples that are annotated at the species taxonomy level are considered, so samples annotated at lower taxonomy levels do not introduce noise into the training of the model.

### 4.2.1. Evaluation Method

Again, a 10-fold cross-validation scheme is applied, so the results show how well the plant species classifier generalizes and do not dependent on a specific training and validation data split. To ensure that multiple temporal cut-outs from a single plant are not present in multiple data splits, the individual plant cut-outs from each species are divided into ten splits based on their plant id. The images are split into 10 folds to test the classification stability. Using these 10 folds, we report the performance of the plant classification algorithm using the average classification accuracy and top-5 recall.

### 4.2.2. Baseline

The baseline classification model is based on the TensorFlow-Slim [44] implementation of ResNet50-v1 [43]. The model parameters are initialized with weights pre-trained on the ImageNet dataset [45]. The model's training parameters are set to the default values as described in TensorFlow's example implementation. The baseline model is evaluated using the above-mentioned evaluation

approach. Each model is trained for 30,000 iterations with a batch-size of 32 on the training data and then evaluated on the evaluation data for each split. The results for the baseline models are summarized in Table 2.

**Table 2.** Results from the plant species classification baseline. The reported values are the mean and standard deviation over all ten splits in the 10-fold cross-validation scheme.

| Metric | Score |
|---|---|
| Accuracy | $77.06 \pm 5.71$ |
| Top 5-recall | $96.79 \pm 1.80$ |

## 5. Discussion

The Open Plant Phenotype Database represents a wide range of different plant species from multiple plant families, which are cultivated under three unique growing conditions. The number of species and the growing conditions ensures a relatively high diversity in the visual appearance of the samples (both inter-species and intra-species), compared to other public datasets in the domain, see Section 1. The 47 species included in the dataset, were primarily selected based on natural occurrence and their importance for arable crop farming in Denmark. However, these species might not be representative for the biodiversity in other countries. Additionally, the applied growing conditions do not induce all potential visual representations of the plant species, as the visual appearance of plants is affected by other factors, including wind, insects, and even dew on leaves. Therefore, observable differences are to be expected when moving into the field or to other countries, since the phenotypic variations of different species change geographically due to the differences in selection pressure and gene pool [19,20]. Still, the three applied growing conditions provide examples of how plants appear when cultivated in two opposite extreme settings (optimum and worst growth condition) and a single field-realistic setting. Although all plant species have been cultivated using the same three growing conditions, it should be noted that the plants have been cultivated over four trials/growth seasons with different weather conditions, which also affects the visual appearance. Additionally, observed big differences in the germination success have been observed across the different species and the different growth conditions, which lead to high variations in number annotated samples in the different groups, which should be taken into account when using the data, see Tables A1 and A2. Furthermore, due to the continuous thinning of the plants during each trial, there will be more representations of early growth stages compared to later growth stages. Thus, the majority of the 64, 292 individual plant will only have been tracked for a few days.

Initially, the bounding box annotation required some effort, as the RoboWeedMaPS' assisting annotation model were only trained on in-situ images, which represent a different environmental setting and plant species distribution. However, the assisting annotation model did in most cases provide decent region proposals, and as more and more OPPD images were manually validated, the robustness of the assisting annotation model was incrementally improved. The most commonly observed challenges for the assisting annotation model were overlapping between multiple plants, far-spaced leaves for a single plant, and tiny cotyledons. These issues were especially challenging for yet unencountered species with unique appearances. In the RoboWeedMaPS online annotation tool, all bounding box annotations were aligned with the image axis. However, the use of axis-aligned annotations were not ideal, when narrow plants were orientated diagonally. Here, rotated bounding boxes might have been a beneficial way of minimizing background clutter as demonstrated by Li et al. [46] for detecting ships in satellite images.

RoboWeedMaPS's correction tool made it convenient to validate and annotate species for multiple images simultaneously. However, it could sometimes be challenging to identify outliers, if the outliers were form the same plant family, or before the plants developed their true leaves, due to highly similar visual traits. To overcome this, the plants were backtracked in time, as they develop more characteristic at later growth stages. However, this approach was not always applicable as the plant continuously

thinned during the trials. All species annotations follow the EPPO encoding (https://gd.eppo.int/), which allows one to easily convert the annotations to other hierarchical levels (e.g., family or genus rather than species) dependent on the application of the data.

The temporal tracking algorithm provided an efficient way link bounding box annotations in time. However, the temporal tracking might not be 100% accurate, as it has been performed using an automated algorithm (e.g., errors might have occurred, if the plants are placed very close to each other). That being said, the algorithm appears quite robust and has not been observed to make any incorrect matching.

Both baseline experiments were conducted using "out-of-the-box" model implementations. Both models do a decent job at their respective tasks, but there are room for improvement. These results further highlight the need for this dataset, as established model designs are insufficient to perform well on the presented applications. The primary challenges observed for the instance detection baseline were overlapping leaves, resulting in multiple plants getting detected as single instances, and plants with large spacing between leaves getting detected as multiple plant instances. Similarly, it can be challenging for the classification baseline to distinguish between different species before the plants develop their first true leaves, or to distinguish between species from the same family or genus (e.g., SONOL, SONAS).

Generally, other established methods for visual recognition of plants have reported higher performance than the baseline experiments [4]. However, these methods are usually based on human-annotated data, thus the results might not necessarily be representative of the real in the field performance, but rather evidence of how good the method is at replicating the human ability to classify plants. Small seedlings are especially difficult to distinguish by a person, and they are therefore often omitted in the training and testing set as they cannot be labeled. With the present dataset, the human ability to recognize weeds is taken out of play, which is reflected in a large number of small plants that would normally be ignored if a person were to annotate them. This is also believed to be one of the explanations for the above classification accuracy, which lies under the accuracy of previous studies including [23].

It should be noted that the primary objective of the baseline experiments was to introduce evaluation methods and metrics, which was achieved for both applications. The baseline experiments both apply a 10-fold cross-validation scheme to make the results less dependent on a specific training and test dataset split. Due to the annotations, it will also be possible to apply exhaustive cross-validation schemes, such as leave-one-out cross-validation (e.g., based on unique box id), however, it would be very computationally heavy.

The dataset is expected to be extended in the future with more plant species, to model more inter-species diversities. Another potential future extension of the dataset is to provide pixel-wise segmentation masks for each plant instance, which would allow for better cut-outs of the individual plants. We would like to invite peers to collaborate and contribute to the databse, for the aim of easier access to high-quality data, from which our field of research can only benefit. OPPD is hosted using Git LFS (https://git-lfs.github.com/), which hopefully will make it easy for peers to contribute to the suggested extensions and to correct any potential errors in the data.

## 6. Conclusions

This work introduced the Open Plant Phenotyping Database, a large-scale public dataset for visual recognition of plants. The dataset is the largest publicly available of its kind, and is expected to a valuable resource for future research in weed detection and plant phenotyping.

The dataset consists of 7590 RGB images, with a bounding box and species annotations for $315,041$ plant objects. The plant objects represent $64,292$ individual plants from 47 different species selected based on natural occurrence in Denmark. The plants have been cultivated using three different growth conditions and tracked temporarily to provide a high degree of intra-species variability with respect to visual appearance of the plants in the collected images.

Finally, the paper presents two experiments/applications of the data: (1) plant instance detection, and (2) plant species classification. The two experiment are used to propose evaluation methods and provide a baseline for future work on the data. The performance of both baselines are non-perfect (plant instance detection: $AP^{IoU=.50:.05:.95} = 37.01 \pm 2.43$, plant species classification: Accuracy $= 77.06 \pm 5.71$), which further illustrates the need for this dataset to boost research in the domain.

**Author Contributions:** Conceptualization, S.L.M. and R.N.J.; Data curation, S.L.M., L.-C.P. and R.N.J.; formal analysis, S.L.M. and M.D.; funding acquisition, S.K.M. and R.N.J.; investigation, S.L.M., S.K.M. and R.N.J.; methodology, S.L.M., S.K.M., M.S.L. and R.N.J.; project administration, S.L.M.; resources, S.K.M. and R.N.J.; software, S.L.M., M.D. and M.S.L.; supervision, S.K.M. and R.N.J.; validation, S.L.M. and S.K.M.; visualization, S.L.M.; writing—original draft, S.L.M.; writing—review and editing, S.L.M., S.K.M., M.D., M.S.L., L.-C.P. and R.N.J. All authors have read and agreed to the published version of the manuscript.

**Funding:** This research was founded by Innovation Fund Denmark as part of the RoboWeedMaPS project (grant number 6150-00027B)

**Acknowledgments:** 

**Conflicts of Interest:** The authors declare no conflict of interests.

## Abbreviations

The following abbreviations are used in this manuscript:

| | |
|---|---|
| OPPD | Open Plant Phenotyping Database |
| DL | Deep Learning |
| EPPO | The European and Mediterranean Plant Protection Organization |
| RGB | Red Green Blue image |
| PPD | Plant Phenotyping Datasets |
| PSD | Plant Seedlings Dataset |
| SBD | Sugar Beets Dataset |
| LCD | Leaf Counting Dataset |
| G1,G2,G3 | Growth Condition 1-3 |
| IoU | Intersection over Union |
| AP | Average Precision |
| AR | Average Recall |
| GIT LFS | GIT Large Files Storage |
| UAV | Unmanned Aerial Vehicle |

## Appendix A. Database Content Details

Tables A1 and A2 show additional details about the distribution of data for full box images and individual plant cut-outs respectively.

**Table A1.** Database content. Number of full box images and bounding boxes for each of the sown/primary species in the dataset. Note some species are cultivated in multiple trial seasons or twice in a single season, to increase germination success.

| Primary Plant Species in Box | English Name | Number of Full Box Images | Number of Bounding Boxes | Trials |
|---|---|---|---|---|
| ALOMY | *Blackgrass* | 141 | 3176 | 2017autumn |
| ANGAR | *Scarlet pimpernel* | 144 | 13,848 | 2017autumn |
| APESV | *Loose silky-bent* | 142 | 3011 | 2017autumn |
| ARTVU | *Common mugwort* | 149 | 5667 | 2017autumn |
| AVEFA | *Common wild oat* | 116 | 3434 | 2017autumn |
| BROST | *Barren brome* | 141 | 2146 | 2017autumn |
| BRSNN | *Rapeseed* | 140 | 4491 | 2017spring |
| CAPBP | *Shepherd's purse* | 149 | 17,566 | 2017spring |
| CENCY | *Cornflower* | 149 | 5409 | 2017spring |
| CHEAL | *Fat-hen* | 305 | 6736 | 2017spring + 2018summer |
| CHYSE | *Corn marigold* | 152 | 1371 | 2018summer |
| CIRAR | *Creeping Thistle* | 256 | 3467 | 2017spring + 2017autumn |
| CONAR | *Field bindweed* | 252 | 1379 | 2018summer + 2019summer |
| EPHHE | *Umbrella milkweed* | 319 | 970 | 2018summer + 2019summer |
| EPHPE | *Stinging milkweed* | 145 | 6186 | 2017autumn |
| EROCI | *Common stork's-bill* | 138 | 4919 | 2017autumn |
| FUMOF | *Common fumitory* | 230 | 552 | 2018summer + 2019summer |
| GALAP | *Cleavers* | 150 | 1788 | 2017autumn |
| GERMO | *Dove's-foot crane's-bill* | 136 | 4771 | 2017spring |
| LAPCO | *Nipplewort* | 140 | 1696 | 2017autumn |
| LOLMU | *Italian ryegrass* | 142 | 3684 | 2017autumn |
| LYCAR | *Common bugloss* | 152 | 478 | 2018summer |
| MATCH | *Scented mayweed* | 154 | 9050 | 2017spring |
| MATIN | *Scentless mayweed* | 150 | 16,060 | 2017spring |
| MELNO | *Night-flowering catchfly* | 146 | 6995 | 2017spring |
| MYOAR | *Field forget-me-not* | 155 | 5083 | 2017spring |
| PAPRH | *Common poppy* | 134 | 24,713 | 2017spring |
| PLALA | *Narrowleaf plantain* | 117 | 4501 | 2017autumn |
| PLAMA | *Broadleaf plantain* | 153 | 5169 | 2018summer |
| POAAN | *Annual bluegrass* | 139 | 9868 | 2017autumn |
| POLAV | *Prostrate knotweed* | 230 | 2659 | 2018summer + 2019summer |
| POLCO | *Black bindweed* | 144 | 2496 | 2017spring |
| POLLA | *Pale smartweed* | 140 | 4034 | 2017spring |
| POLPE | *Redshank* | 120 | 3278 | 2017autumn |
| RUMCR | *Curly dock* | 140 | 6672 | 2017autumn |
| SENVU | *Common groundsel* | 152 | 6709 | 2018summer |
| SINAR | *Charlock* | 139 | 4602 | 2017spring |
| SOLNI | *Black nightshade* | 146 | 6823 | 2017spring |
| SONOL | *Common sowthistle* | 193 | 9301 | 2017autumn + 2019summer |
| STEME | *Common chickweed* | 149 | 10,159 | 2017spring |
| THLAR | *Field penny-cress* | 146 | 6506 | 2018summer |
| URTUR | *Small nettle* | 143 | 14,007 | 2017spring |
| VERAR | *Corn speedwell* | 272 | 37,119 | 2×2017autumn |
| VERPE | *Common field speedwell* | 256 | 11,008 | 2×2017spring |
| VICHI | *Common hairy tare* | 147 | 4233 | 2017autumn |
| VIOAR | *Field pansy* | 137 | 7251 | 2017spring |
| Total | | 7590 | 315,038 | |

**Table A2.** Database content. Number of individual plant cut-outs and unique plants for each species in the dataset. The monocotyledons (PPPMM) and dicotyledon (PPPDD) categories cover cut-outs only annotated at these levels. The "Other" category cover cut-outs annotated at lower taxonomic levels than species.

| Species (EPPO) | English Name | Number of Image Cut-Outs | Number of Unique Plants |
|---|---|---|---|
| ALOMY | *Blackgrass* | 2583 | 730 |
| ANGAR | *Scarlet pimpernel* | 10,691 | 790 |
| APESV | *Loose silky-bent* | 2709 | 412 |
| ARTVU | *Common mugwort* | 5360 | 820 |
| AVEFA | *Common wild oat* | 3282 | 1630 |
| BROST | *Barren brome* | 1735 | 537 |
| BRSNN | *Rapeseed* | 4051 | 576 |
| CAPBP | *Shepherd's purse* | 16,411 | 3382 |
| CENCY | *Cornflower* | 4145 | 508 |
| CHEAL | *Fat-hen* | 6321 | 1433 |
| CHYSE | *Corn marigold* | 1324 | 223 |
| CIRAR | *Creeping Thistle* | 1525 | 97 |
| CONAR | *Field bindweed* | 1259 | 71 |
| EPHHE | *Umbrella milkweed* | 173 | 16 |
| EPHPE | *Stinging milkweed* | 5758 | 506 |
| EROCI | *Common stork's-bill* | 4295 | 604 |
| FUMOF | *Common fumitory* | 143 | 39 |
| GALAP | *Cleavers* | 1557 | 217 |
| GERMO | *Dove's-foot crane's-bill* | 3632 | 342 |
| LAPCO | *Nipplewort* | 1349 | 74 |
| LOLMU | *Italian ryegrass* | 3571 | 1309 |
| LYCAR | *Common bugloss* | 381 | 41 |
| MATCH | *Scented mayweed* | 7965 | 1385 |
| MATIN | *Scentless mayweed* | 15,065 | 3490 |
| MELNO | *Night-flowering catchfly* | 5677 | 516 |
| MYOAR | *Field forget-me-not* | 3222 | 323 |
| PAPRH | *Common poppy* | 23,302 | 7351 |
| PLALA | *Narrowleaf plantain* | 3593 | 1065 |
| PLAMA | *Broadleaf plantain* | 4868 | 677 |
| POAAN | *Annual bluegrass* | 9329 | 3134 |
| POLAV | *Prostrate knotweed* | 2411 | 274 |
| POLCO | *Black bindweed* | 1608 | 232 |
| POLLA | *Pale smartweed* | 3393 | 300 |
| POLPE | *Redshank* | 3009 | 421 |
| RUMCR | *Curly dock* | 6264 | 1118 |
| SENVU | *Common groundsel* | 6571 | 1609 |
| SINAR | *Charlock* | 4030 | 588 |
| SOLNI | *Black nightshade* | 6230 | 874 |
| SONAS | *Spiny sowthistle* | 587 | 59 |
| SONOL | *Common sowthistle* | 4646 | 576 |
| STEME | *Common chickweed* | 8612 | 1572 |
| THLAR | *Field penny-cress* | 6369 | 1340 |
| URTUR | *Small nettle* | 7753 | 681 |
| VERAR | *Corn speedwell* | 28,791 | 5338 |
| VERPE | *Common field speedwell* | 9191 | 746 |
| VICHI | *Common hairy tare* | 3982 | 489 |
| VIOAR | *Field pansy* | 6176 | 382 |
| PPPMM | *Monocotyledonous plants* | 9365 | 4006 |
| PPPDD | *Dicotyledonous plants* | 37,990 | 15,366 |
| Other | | 2784 | 787 |
| Total | | 315,038 | 64,292 |

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
