# Peer review of "Open Plant Phenotype Database of Common Weeds in Denmark"

_remotesensing, doi:10.3390/rs12081246_

Round 1
Reviewer 1 Report
The work presents a public annotated dataset for plant detection and plant classification: the Open Plant Phenotype Database (OPPD). This dataset contains 4042 RGB images of 26 plant species. Each species is cultivated under three different growth conditions. These experiments introduce evaluation metrics and methods for plant instance detection and plant species classification, providing baselines for future work on the data. Finally, this paper addresses an important question, which keeps growing in importance.
However, in the description of experimental research, including conclusions, it misses details of the general approach and its applicability. I would propose a more detailed methodology description and a more critical result discussion.
Major comments
Introduction
Phenotyping has become an increasingly important topic in recent years. The provision of this large database could bring a contribution in terms of targeted execution of plant visual assessments improving the effective crop needs and the soil characteristics. Enhance and contextualize this topic with your aims in the introduction section.
Insert some examples of deep learning application for visual recognition of plants.
Introduce something about proximal sensing (this work is an application of this methodology) compared to remote sensing, underlining all the advantages that this technique can provide.
The aim of this work is confused and not very explicit. Expand and define it better.
Methods
Line 78: I don’t think this introductory phrase to methods is necessary as it is easy to understand from the section title.
Section 2.1.2: there are methods to calibrate the images that could improve the output of this work (see Menesatti et al. 2012). Cite these methods to provide a database that is as generalizable as possible.
Section 2.2: there are many studies in literature that combines shape analysis and colorimetric k-nearest neighbor (k-NN) clustering for in-field weed discrimination (see Pallottino et al., 2018) that automatically carry out the detection and not manually. Justify why this phase was done manually and write the benefits.
Discussion
In general, the discussion is poorly organized: I would go step by step with how the results were presented contextualizing them with what is present in the literature.
Also in this section justify why you talk about since the only mentioned phenotyping activity is the visual appearance. Moreover, introduce works in literature that may provide a comparison.
Conclusions
L314-316: I don’t think this is a sentence that should be included in the conclusions of an international scientific article.
Reviewer 2 Report
This article and reference dataset is very meaningful.
But I have a question for author. Is it applicable to special vegetation? such as rice paddy, rubber plantations, oil palm.
Reviewer 3 Report
The authors aimed to create a large annotated RGB image dataset of plants as a common testbed for plant detection and classification algorithms. Such datasets will always make a positive contribution to the research community. The manuscript well documented the dataset in various aspects. Below are my major and minor comments.
Major:
Considering the existence of the datasets mentioned in section 1.1, what is the necessity of building a new dataset like yours? What makes your dataset standing out from the existing ones? What unique functions does your dataset have? These questions are not clearly explained in the manuscript.
Section 2.2 should be written in past tense. Many sentences in section 5 should also use past tense.
Conclusion reads like an abstract. I recommend rewriting it and keeping it concise.
Minor:
Line 33, are there any other public datasets besides the ones mentioned in section 1.1? Since you stated only a few exist, perhaps you can list all of them exhaustively in a table.
Line 54, I only see a few images following your GitLab link.
Line 63, change “On” to “In” or “For”.
Line 89, add a comma after “surface”.
Line 106, it would be a good idea to add a figure showing the imaging system setup.
Line 155, considering your imaging system setup, affine transformation doesn’t seem to be necessary. 2D Euclidean transformation should be sufficient.
Line 159, change “figure 3” to “Figure 3”.
Move Figure 1, 2 and 3 after the paragraphs where you cited them.
Round 2
Reviewer 1 Report
The manuscript in the present form have been significantly improved. Parts that better contextualize the work have been added (e.g. phenotyping, deep learning, etc.). The methodology has been better specified and both aim and discussion are now clearer.
Author Response
Thank you for your valuable feedback.
On behalf of the authors,
Simon Leminen Madsen
Reviewer 3 Report
Sufficient changes have been made to the manuscript.
Author Response

(The authors gave the same response as above.)
